# Unraveling the transcriptomic effects of leucine supplementation on muscle growth and performance in basketball athletes

Sinan Wang[1], Weishuai Guo[2,3], Rui Dong[4]*

1 College of Arts & Physical Education, Sejong University, Seoul, Korea, 2 Department of Physical Education, Jeonbuk National University, Jeonju, Korea, 3 Department of Physical Education, Pingdingshan University, Pingdingshan, China, 4 School of Physical Education and Sports Science, South China Normal University, Guangzhou, China

* dongrui@scnu.edu.cn

**Data Availability Statement:** All the data are contained within the manuscript and supporting information file.

## Abstract

Leucine has gained recognition as an athletic dietary supplement in recent years due to its various benefits; however, the underlying molecular mechanisms remain unclear. In this study, 20 basketball players were recruited and randomly assigned to two groups. Baseline exercise performance—assessed through a 282-foot sprint, free throws, three-point field goals, and self-rated practice assessments—was measured prior to leucine supplementation. Participants were then given a functional drink containing either leucine (50 mg/kg body weight) or a placebo for 28 days. After supplementation, the same exercise performance metrics were reassessed. Following leucine supplementation, biceps brachii muscle tissue from both groups was collected for transcriptome sequencing and qPCR verification. Our results suggested that leucine supplementation significantly improved 282-foot sprint performance, reducing times from 17.4 ± 0.9 to 16.2 ± 0.9 seconds in the leucine group, compared to minimal changes in the control group (from 17.3 ± 0.9 to 17.1 ± 0.8 seconds; P = 0.034). For other exercise performance metrics, no significant differences were observed (P > 0.05); however, trends toward improvement were noted. Transcriptomic analysis revealed 3,658 differentially expressed genes (DEGs) between the two groups. These DEGs were enriched in pathways related to immune response (P < 0.0001), positive regulation of cytokine production (P < 0.0001), and neutrophil extracellular trap formation (P < 0.0001), among others. Weighted Gene Co-expression Network Analysis (WGCNA) identified a module (turquoise) strongly associated with muscle growth, with DEGs in this module enriched in cytoskeletal pathways in muscle cells. Gene expression changes (α-tubulin, β-tubulin, CK18, CK8, vimentin, cofilin, gelsolin, profilin, MAP1, MAP2, MAP4, E-cadherin, and N-cadherin) were verified by qPCR. In summary, leucine supplementation improved exercise performance, particularly by significantly reducing sprint times and showing trends of improvement in other performance metrics, including three-point field goals, free throws, and self-rated well-being. Identified DEGs enriched in pathways related to immune response, cytokine production, and cell adhesion. WGCNA highlighted a key module associated with muscle growth, enriched in cytoskeletal pathways. qPCR validation confirmed the upregulation of cytoskeleton-related genes, supporting the transcriptomic findings.

**Funding:** The author(s) received no specific funding for this work.

**Competing interests:** The authors have declared that no competing interests exist.

These results suggest that leucine enhances muscle adaptation by regulating cytoskeletal dynamics, providing molecular insights into its role in improving athletic performance.

## 1. Introduction

Among the branched-chain amino acid (BCAA), leucine is considered a key amino acid in exercise performance and muscle function, especially in athletic activities like basketball [1–3]. Like many amino acids, leucine is an essential amino acid integrated in protein synthesis and muscle repair [4]. Many trainings have determined that leucine has the ability to stimulate muscle protein synthesis through the induction of the activation of mammalian target of rapamycin (mTOR), which is a key determinant of cell growth and metabolism [5]. Such induction very much encourages the hypertrophy and strength of muscles, which are necessary in basketball, given the characteristic activities involved in fixed, high-intensity, short-duration bursts [6]. Further, amino acid leucine has been found to enhance post-exercise muscle recovery by reducing muscle soreness and damage, something rather important to highly trained athletes [7, 8]. Apart from its anabolic action, leucine can help provide energy supply during exercises through gluconeogenesis and, hence, maintain glucose homeostasis and prolong the development of fatigue [9]. Hence, supplementation with leucine would result in increased endurance, power output, and overall athletic performance [10].

The effects of leucine on muscle function and exercise performance are multi-faceted; therefore, deeper explorations are required for an explanation of the molecular mechanisms involved. While leucine is well-acknowledged to stimulate muscle protein synthesis to support recovery, the precise pathways through which it enhances athletic performance in dynamic sports like basketball have yet to be fully elicited. Comprehensive transcriptomic analyses provide such valuable insights with the unmasking of gene expressions changed due to leucine supplementation that would explain its function in optimizing muscle health and performance. The genes of interest were selected based on their critical roles in maintaining cytoskeletal integrity, regulating muscle contraction, and promoting tissue repair, all of which are essential for muscle adaptation to physical stress and high-intensity activity. Specifically, *α-tubulin* and *β-tubulin* are key components of microtubules, which provide structural support and facilitate intracellular transport [11]. *CK18* and *CK8*, as intermediate filaments, contribute to cellular integrity and resilience under mechanical stress [12]. *Vimentin* is involved in maintaining cell shape and stability during dynamic cellular processes [13]. Actin-binding proteins, such as *cofilin*, *gelsolin*, and *profilin*, regulate actin filament assembly and disassembly, crucial for muscle contraction and cytoskeletal remodeling [14]. *MAP1*, *MAP2*, and *MAP4* are microtubule-associated proteins that modulate microtubule stability and dynamics, supporting cellular structure during muscle activity [15]. *E-cadherin* and *N-cadherin* are adhesion proteins that play pivotal roles in cell-cell adhesion, necessary for tissue integrity and coordinated muscle function [16]. These genes collectively represent fundamental processes such as actin and microtubule dynamics, cell adhesion, and cytoskeletal stability, which leucine supplementation may influence to enhance muscle strength, function, and recovery. Once these molecular pathways are elucidated, much deeper insight into the potential for leucine to optimize performance in sports requiring explosive power and endurance will be acquired.

Basketball is a high-intensity, intermittent sport that requires a combination of endurance, strength, power, agility, and skill execution [17]. Therefore, the necessity and innovation of researching leucine supplementation specifically in high-level basketball athletes are

particularly compelling. The high-level basketball players must undergo serious training schedules and competitive demands quite different from other sports [18]. Their training regimens require a precise balance of strength, agility, and endurance [19, 20]; thus, this would be an ideal population in which to study the specific impacts of leucine supplementation. The selected performance variables—282-foot sprint, free throws, and three-point field goals—were chosen based on their relevance to the physiological and skill demands of basketball. The sprint test was designed to assess anaerobic capacity, which is critical for high-intensity, short-duration efforts, as well as the activation and performance of fast-twitch muscle fibers required for rapid and explosive movements during gameplay [21, 22]. Free throws and three-point shots were included to evaluate precision, muscle coordination, and endurance [23, 24], all of which are necessary for maintaining shooting accuracy under physical and mental fatigue. Given the unique physiological and metabolic demands put on elite basketball players, specialized nutritional strategies, such as leucine supplementation, could provide new insights into ways to enhance performance and recovery.

While the general benefits of leucine for athletic performance are well-documented, the specific effects in basketball remain under-researched, particularly at the level of how leucine supplementation affects key performance metrics, recovery, and muscle function. Innovative research focusing on this population could lead to the development of sport-specific nutritional protocols that enhance recovery, reduce injury risk, and improve overall performance. Such research not only has the potential to advance the scientific understanding of leucine's role in muscle metabolism but also to provide practical, evidence-based recommendations for basketball athletes and their coaches.

## 2. Materials and methods

### 2.1 Participant demographics

A total of 20 male basketball players, aged 18–23 years, were recruited from a university basketball team to participate in this study. All participants possessed a Level 1 basketball athlete qualification. Recruitment was conducted through advertisements, and the study was approved by the university's ethics committee. Table 1 lists subjects' demographics and demonstrates no statistically significant difference between control group and experimental group. Participants' body weight and height were measured to ensure comparability between the two groups, and no significant differences were detected. The experiment took place from 13

**Table 1. Participant demographics.**

|  | control (N = 10) | luecine supplementation group (N = 10) | P value |
|---|---|---|---|
| Height (m) | 1.93 ± 0.05 | 1.96 ± 0.07 | 0.34 |
| Range | 1.87–2.04 | 1.87–2.11 |  |
| Weight (kg) | 90.1 ± 2.8 | 88.9± 2.5 | 0.35 |
| Range | 85.6–95.1 | 84.4–92.1 |  |
| BMI (kg/m2) | 24.3 ± 1.5 | 23.3 ± 1.6 | 0.19 |
| Range | 21.4–27.2 | 20.2–25.1 |  |
| Position on team |  |  |  |
| Guard n (%) | 7.0 (70.0) | 6.0 (60.0) |  |
| Forward n (%) | 2.0 (20.0) | 1.0 (10.0) |  |
| Center n (%) | 1.0 (10.0) | 3.0 (30.0) |  |

Data presented as mean ± standard deviation unless otherwise indicated; BMI (body mass index).

December 2023 to 4 February 2024, and written informed consent was obtained from all participants before the study commenced. This experiment has been approved by the Ethics Committee of South China Normal University (SCNU-SPT-2023-031).

## 2.2 Study design

Before the experiment, we measure their exercise performance like 282 feet sprint, free throws, three-point field goals and subject self-rating at practices as baseline, then we provided them functional drink with or without leucine supplementation before and after daily training for 28 days. Subsequently, the same exercise performances were recorded as above. During the experimental period, the athletes will have four hours of closed training every day, and make sure they have ten hours of sleep a day.

**2.2.1 Controlled diet.** Participants adhered to a controlled diet during the exercise test, consisting of prepackaged meals and snacks designed to provide 1.5 times their resting energy expenditure (REE). The diet supplied 1.2 grams of protein per kilogram of body weight per day (g/kg/d), 4 g/kg/d of carbohydrates, and the remaining calories from fat to ensure a balanced intake aligned with their energy needs. Participants consumed their last prepackaged meal approximately 2–3 hours before each exercise test to ensure adequate digestion and energy availability while minimizing the risk of gastrointestinal discomfort during performance assessments. This timing was standardized across all participants to maintain consistency. To maintain consistency in hydration and nutrient intake, participants were instructed to drink only water in addition to the provided meals and snacks, minimizing variables that could impact exercise performance.

**2.2.2 Functional drink composition.** Under researcher supervision, all participants were given a $0.25 \text{ g·kg}^{-1}$ carbohydrate solution mixed with 16 ounces of water (total $0.50 \text{ g·kg}^{-1}$) 30 minutes before and immediately after exercise. The independent variable was the supplementation of $25 \text{ mg·kg}^{-1}$ of leucine powder added to the test drinks provided both before and after exercise, yielding a total dosage of $50 \text{ mg·kg}^{-1}$ for participants in the experimental group. Blind taste tests determined that the carbohydrate solution mixed in 16 ounces of water was enough to dilute the taste of leucine and that the taste of the drinks was indistinguishable.

## 2.3 Athletic performance measures and testing

Specific measures of basketball performance were measured before and after the 28-day experimental period to assess changes in performance. Training sessions were usually conducted in the afternoon, and performance measures were usually recorded between 14:00–18:00. Performance assessments consisted of timed sprints and shooting accuracy. The first measure involved a 282-foot timed sprint, requiring participants to run from the baseline to the half court and back, followed by a sprint to the full court and back to the baseline. To ensure consistency, the same individual recorded the sprint times after each session. The second measure evaluated free-throw shooting accuracy, with participants attempting 10 free throws from 15 feet. The number of successful attempts was recorded to assess accuracy. The third measure focused on three-point shooting accuracy. Participants performed 15 three-pointers in three sets of five shots each. The first set was from the right corner, the second from the top of the key, and the third from the left corner of the court. The number of successful shots in each set was recorded. After each session, participants rated their mental and physical wellbeing on a 10-point scale, which gave subjective information regarding their condition and preparedness during exercise. The sample size for this study was determined using a power analysis to ensure sufficient statistical power for detecting changes in performance metrics (e.g., sprint

times, free-throw accuracy) between the experimental and control groups. A two-tailed analysis with a power of 80% ($\beta = 0.2$) and a significance level of 0.05 ($\alpha = 0.05$) was conducted.

## 2.4 Transcriptomic experimental design

At the end of the 28-day experimental period, muscle samples from the biceps were collected from each group for RNA sequencing (RNA-seq) analysis. There were six replicate muscle samples for each group. The muscle samples of the biceps were excised aseptically and immediately flash-frozen in liquid nitrogen for the preservation of RNA. They were then kept at -80˚C until RNA extraction, ensuring that the samples remained suitable for high-quality RNA sequencing analysis. The raw data of RNA-seq can be found in the S1 File.

## 2.5 Data processing and differential expression analysis

Mapping of official gene symbols to commercial probe IDs was the first step in processing the microarray data. The data was then uploaded to NetworkAnalyst 3.0 (https://www. networkanalyst.ca) for further processing. We excluded genes with low variance (percentile rank below 15), low abundance (below 4), or no annotation. Next, we applied a log2 transformation to normalize gene expression levels. Using the Limma package, we identified genes that were significantly upregulated or downregulated. The Benjamini–Hochberg method was applied to correct for multiple testing, setting the threshold for differentially expressed genes (DEGs) at an absolute log2 fold change greater than 0.5 and an adjusted p-value below 0.05.

## 2.6 Functional enrichment analysis

We employed g:Profiler (http://biit.cs.ut.ee/gprofiler/) to perform GO and KEGG path-way enrichment analyses on the DEGs. The g:GOSt tool within g:Profiler, along with the g:SCS algorithm, was used to determine significant enrichments, with a significance threshold of $p < 0.05$. The GO analysis encompassed Biological Processes (BP), Cellular Components (CC), and Molecular Functions (MF).

## 2.7 Protein-Protein Interaction (PPI) network construction

Total RNA was isolated using the RNeasy mini kit (Qiagen). Strand-specific RNA-seq libraries were prepared with the TruSeq stranded total RNA sample preparation kit (Illumina), quantified by the Qubit 2.0 Fluorometer (Life Technologies), and assessed for insert size with the 2100 bioanalyzer (Agilent). Clusters were generated on the cBot at a concentration of 10 pM and sequenced on the NovaSeq 6000 system (Illumina). Library preparation and sequencing were carried out by the Meiji Biotechnology Corporation.

## 2.8 Weighted Gene Co-Expression Network Analysis (WGCNA)

We constructed an unsigned co-expression network using the WGCNA package in R. Genes with similar expression patterns were clustered together based on topological overlap, and the Dynamic Hybrid Tree Cut algorithm was used to define modules within the clustering tree. We set a minimum module size of 30 genes and a minimum height for merging modules at 0.25. Modules were assigned random colors for identification. We summarized each module by its module eigengene, which is the first principal com-ponent of the module's expression profiles. These module eigengenes were then correlated with clinical traits such as muscle growth.

## 2.9 RNA isolation and Q-PCR assay

Total RNA was extracted using TRIzol reagent (Thermo) as per the manufacturer's instructions. Quantitative PCR was conducted with SYBR Green reagent (Takara) on a LightCycler 480 system (Roche). Expression levels were normalized to human ACTB.

## 2.10 Statistical analysis

We analyzed the data using GraphPad software, Prism 8.0.1. We performed two-way ANOVA, evaluating the impact of two independent variables and their interaction on the dependent variable. Further, we conducted Tukey's multiple comparison test to find differences among groups. A P-value < 0.05 was considered significant.

# 3. Results

## 3.1. Leucine supplementation promotes basketball athletes' exercise performance

Improvement was observed in the 282-foot sprint (Fig 1A). Sprint time significantly decreased from 17.4 ± 0.9 to 16.2 ± 0.9 seconds after leucine supplementation, compared to the control group, which showed a minimal change from 17.3 ± 0.9 to 17.1 ± 0.8 seconds (P = 0.034). For other exercise performances, three-point field goals increased from 10.2 ± 1.1 to 11.1 ± 1.4 after leucine supplementation, compared to the control group, which changed from 10.5 ± 1.0 to 10.4 ± 0.8 (Fig 1B). Free throws increased from 8.0 ± 0.8 to 8.4 ± 0.8 after leucine supplementation, compared to the control group, which increased slightly from 7.8 ± 1.2 to 7.9 ± 0.7 (Fig 1C). However, no significant differences were observed in these metrics (P > 0.05), although a trend toward improvement was noted.

The subjective ratings at baseline revealed no differences in mental and physical well-being during practice. However, after the 28-day experimental period, leucine supplementation appeared to promote these subjective feelings, with scores increasing from 7.3 ± 0.9 to 7.9 ± 0.7 in the leucine group, compared to a slight increase from 7.1 ± 0.9 to 7.3 ± 0.9 in the control group (Fig 1D). Nevertheless, this improvement was not statistically significant (P > 0.05).

## 3.2. Data preprocessing and Identification of Differentially Expressed Genes (DEGs)

To investigate the behind molecular mechanism, we collected the biceps brachii muscle tissue from two groups after leucine supplementation for transcriptome sequencing. The PCA analysis showed that leucine supplementation can significantly change the transcriptomic profile in muscle, each point represents one sample, and they can be divided into 2 groups as treatment (Fig 2A). Through pairwise comparisons of muscle samples from two groups, a total of 3658 genes were identified in terms of log2(fold change) ≥ 2 and p_adj < 0.05. Specifically, there were 2142 DEGs were down-regulated and 1516 DEGs were up-regulated in leucine supplementation group (Fig 2B). The volcano plot revealed a significant up-regulation of *CYP1A1*, *KCNE5*, *VWA3B*, *SPP1*, *BMPR2*, *SPON1*, *BMPR1B*, *ADAM12*, *BMPR1A*, *TNC*, *SMAD7* with higher fold changes observed after leucine supplementation, whereas *EMSLR*, *GOLGA6GP*, *MTCYBP18*, *ERICD*, *CALML6*, *CA3-AS1* and *CCL18* exhibited decreased expression levels (Fig 2C).

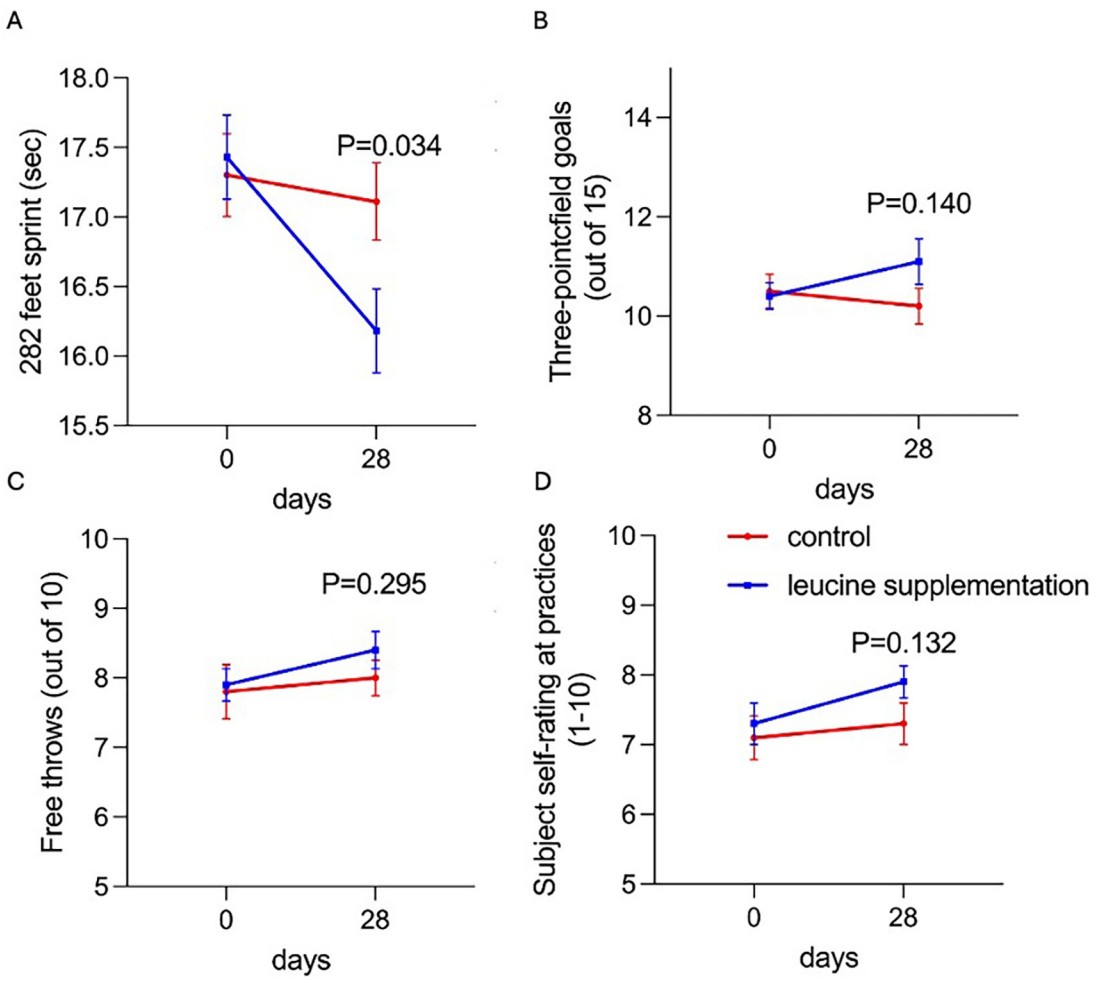

**Fig 1. Leucine supplementations promote basketball athlete's exercise performance.** Indices of athletic performance specific to basketball were measured before experiment and after 28 days experimental period to assess changes in performance. (A) The 282 feet sprint (baseline to half-court and back to baseline, then to full-court and back to baseline) and was timed after each practice by the same person. The second and third performance indices were (B) free throw and (C) 3-point shooting accuracy. (D) In addition, subjects' subjective mental and physical well-being were assessed after every practice by soliciting how they felt during the practice on a 10-point rating scale.

## 3.3. Functional analysis of the DEGs

The function of DEGs was explored using GO enrichment analysis (Fig 3A). The top 5 significance terms were activation of immune response (p.adj = $1.82 \times 10^{-18}$), positive regulation of cytokine production (p.adj = $1.82 \times 10^{-18}$), leukocyte cell-cell adhesion (p.adj = $2.85 \times 10^{-17}$), regulation of cell-cell adhesion (p.adj = $3.42 \times 10^{-17}$) and immune response regulating signaling pathway (p.adj = $5.44 \times 10^{-17}$). The KEGG pathways with significant enrichment are presented in Fig 3B, including Neutrophil extracellular trap formation (p.adj = $7.54 \times 10^{-18}$), Systemic lupus erythematosus (p.adj = $1.67 \times 10^{-17}$), Complement and coagulation cascades (p.adj = $4.06 \times 10^{-9}$), Osteoclast differentiation (p.adj = $7.46 \times 10^{-9}$) and Alcoholism (p. adj = $3.57 \times 10^{-6}$).

Beyond the GO and KEGG pathway enrichment analysis of differentially expressed genes (DEGs), we conducted a Gene Set Enrichment Analysis (GSEA) focused on Gene Ontology (GO) categories across all quantified genes. This extensive analysis allowed us to pinpoint the

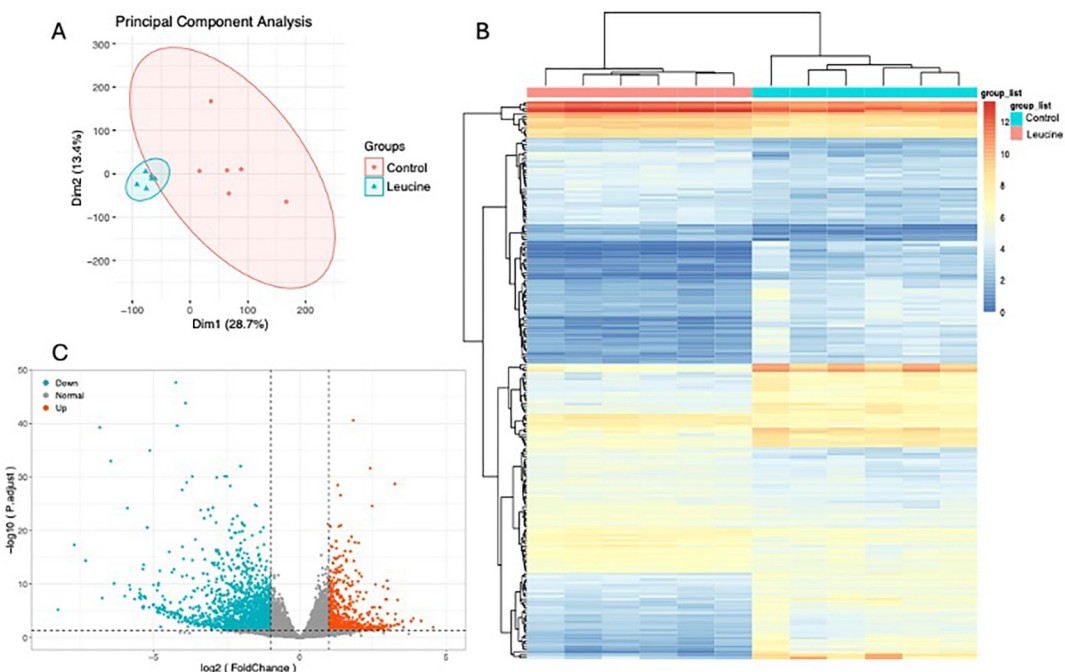

**Fig 2. Data preprocessing and identification of Differentially Expressed Genes (DEGs).** (A). PCA analysis of 2 groups (n = 6). (B). The heatmap of DEGs between 2 groups. (C) The volcano map of the DEGs, left mean down-regulated, right means up-regulated.

gene sets that were most significantly overrepresented in our dataset, shedding light on the cellular mechanisms and pathways that play a pivotal role in the cellular functions being studied. Notably, our findings indicated that the signaling pathway related to Angiogenesis was significantly enriched following leucine supplementation (Fig 3C).

Then, we performed protein-protein interaction analysis, want to figure out the central protein interaction and identified key DEGs, the results showed *ACVR2B*, *SMAD3*, *NODAL*, *SMAD2*, *SMAD7*, *ACR2A*, *SMAD4*, *BMRP1A*, *ACVR1*, *BMPR1B*, *BMRP2*, *TGFBR2* and *TGFBR1* are top 13 hub genes (Fig 3D).

## 3.4. Weighted Gene Co-Expression Network Analysis (WGCNA) identifies critical modules correlating with muscle growth

Transcriptomic analysis was conducted on each micro dissected tubule specimen. While previous genome-wide expression studies have primarily focused on the associations following leucine supplementation, our approach involved an impartial analysis of gene expression to uncover groups of co-expressed genes and their respective modules within our dataset. Weighted Gene Co-expression Network Analysis (WGCNA) is a method rooted in systems biology that is employed to decipher the complex patterns of correlation among genes across various samples. WGCNA enables the identification of gene clusters, known as modules. Each module is characterized by its eigengene, providing a concise summary of the module's overall gene expression profile. The relationship between these eigengenes and sample characteristics, including clinical parameters, can then be explored to gain deeper insights into the biological significance of these gene networks.

Hierarchical clustering analysis (HCA) demonstrated that all six replicates within each group clustered consistently, without any discernible outliers, underscoring the high

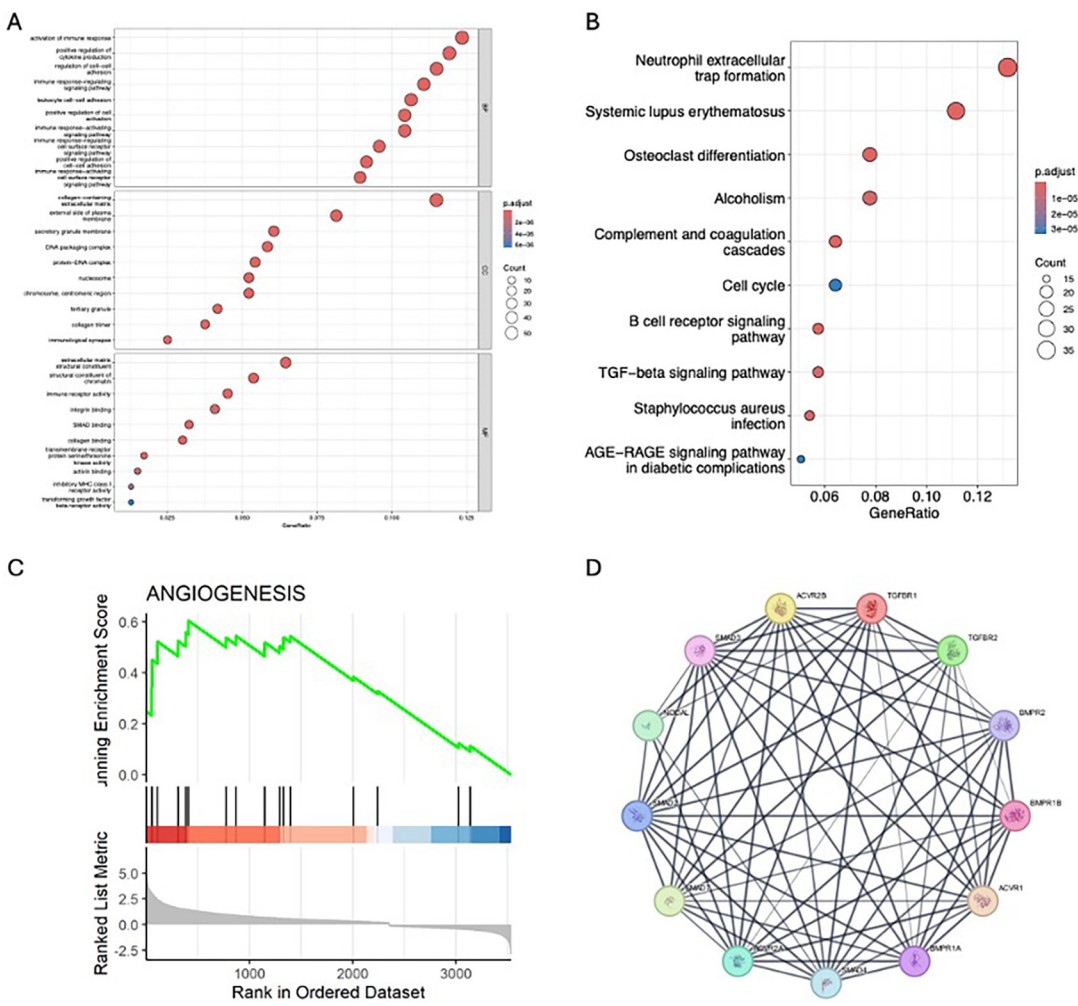

**Fig 3. Functional analysis of the DEGs.** (A) The GO analysis of the DEGS, (B) The KEGG analysis of the DEGs, (C) The GSEA analysis of the total identified genes, (D) the protein-protein interaction analysis of the DEGs.

reproducibility among samples (Fig 4A). Subsequently, Weighted Gene Co-expression Network Analysis (WGCNA) was employed on the Differentially Ex-pressed Genes (DEGs) identified earlier, resulting in the clustering of 5000 genes into modules delineated by distinct colors based on their expression profiles. Notably, the turquoise module emerged as the predominant module (Fig 4C).

To ascertain the relationship between modules and traits, we utilized the Pearson correlation coefficient (r > 0.5) and evaluated significance through p-values (p < 0.05). Our analysis revealed a robust correlation between the turquoise module and the trait of interest (r = 0.93; p = 2e-05), thus prompting us to focus further investigations on this module (Fig 4B).

In the subsequent network heatmap plot (Fig 4D), depicting gene co-expression interconnectivity alongside hierarchical clustering dendrograms and resultant modules, the intensity of yellow and red colors signifies high co-expression interconnectedness, consistent with the dominance of the turquoise module observed in our study.

Fig 4E showcases a heatmap specifically highlighting genes from the turquoise module, illustrating distinct differences in expression patterns across the two experimental conditions.

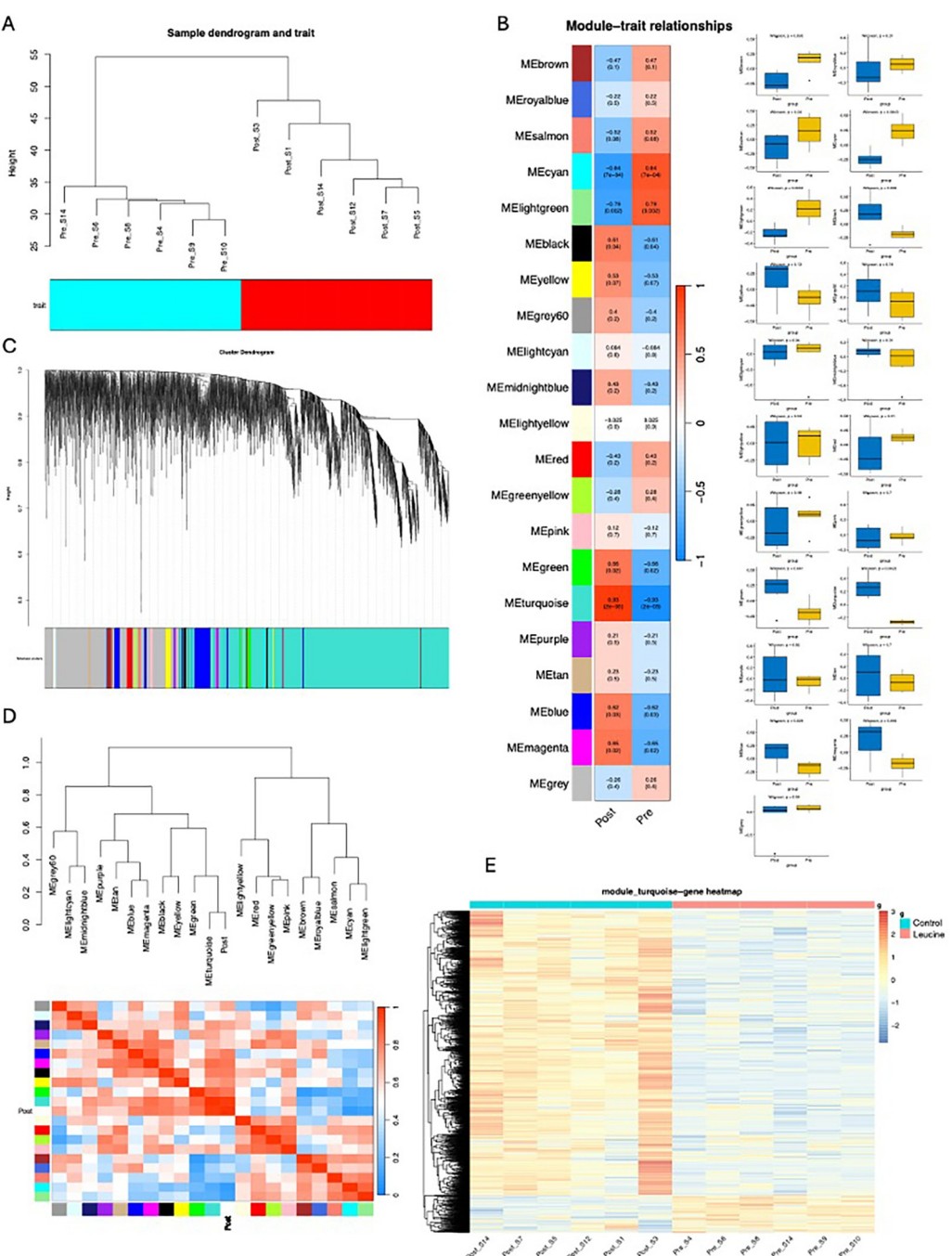

**Fig 4. Weighted Gene Co-Expression Network Analysis (WGCNA) identifies critical modules correlating with muscle growth.** (A), Hierarchical clustering analysis (B), To ascertain the relationship between modules and traits, we utilized the Pearson correlation coefficient (r > 0.5) and evaluated significance through p-values (p < 0.05) (C), resulting in the clustering of 5000 genes into modules delineated by distinct colors based on their expression profiles. (D), network heatmap plot (E), a heatmap specifically highlighting genes from the turquoise module.

## 3.5. Functional analysis of the turquoise module genes

The function of DEGs was explored using GO enrichment analysis (Fig 5A) he KEGG pathways with significant enrichment are presented in Fig 5B. The cytoskeleton in muscle cells

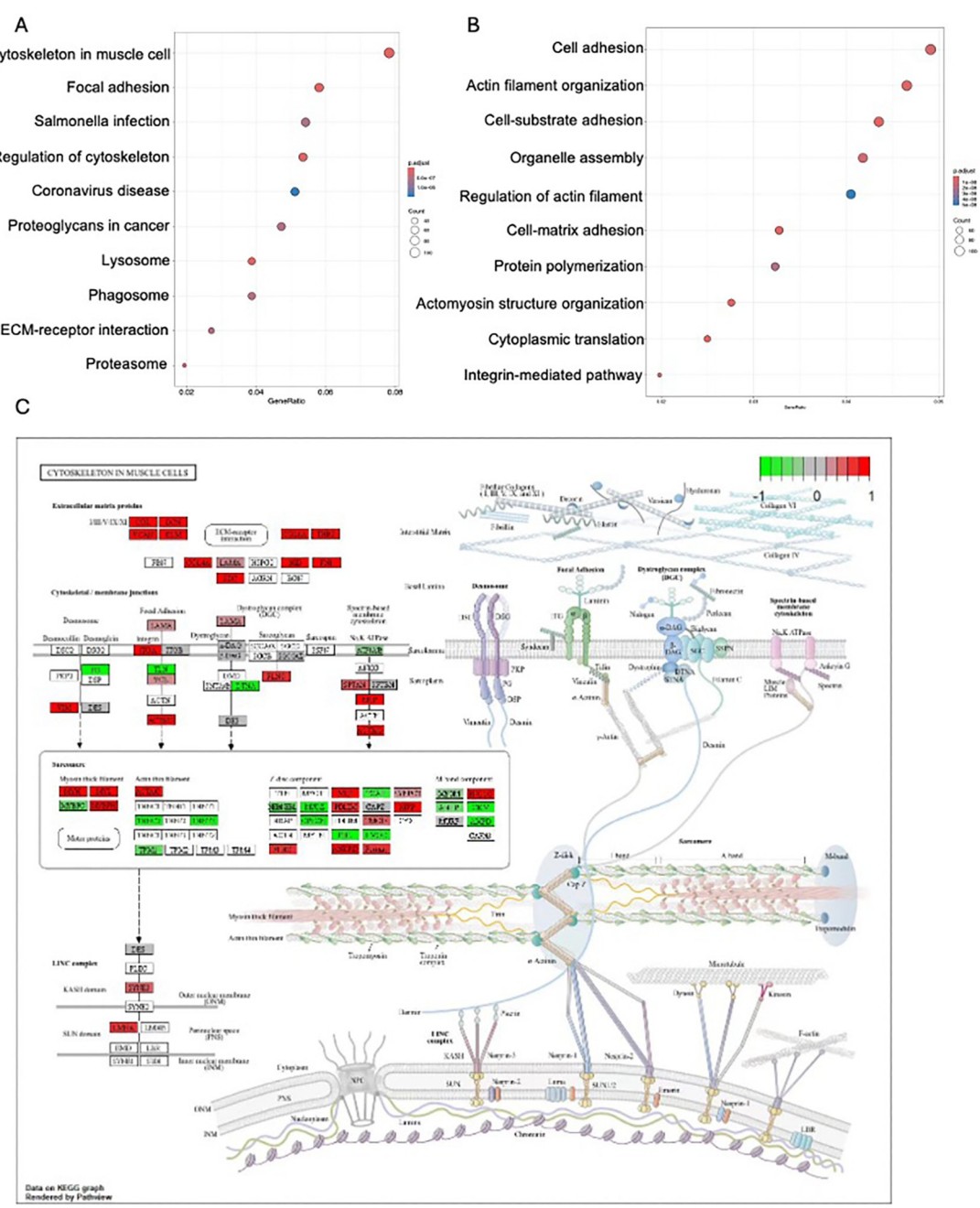

**Fig 5. Functional analysis of the turquoise module genes.** (A) The GO analysis of turquoise model genes, (B) The KEGG analysis of turquoise model genes, (C) The genes involved in cytoskeleton in muscle cells pathway, red means up-regulated, yellow means down-regulated.

plays a crucial role in muscle growth, also known as hypertrophy, as well as in the maintenance of muscle structure and function [25, 26]. The cytoskeleton is a network of protein fibers within the cell that provides structural support, organizes cellular com-ponents, and facilitates cell signaling, division, and movement [27]. Our results indicated that DEGs were enriched on cytoskeleton (Fig 5C). The cytoskeleton maintains the structural integrity of muscle cells (myocytes) and ensures the proper alignment of myofibrils, which are the contractile units of

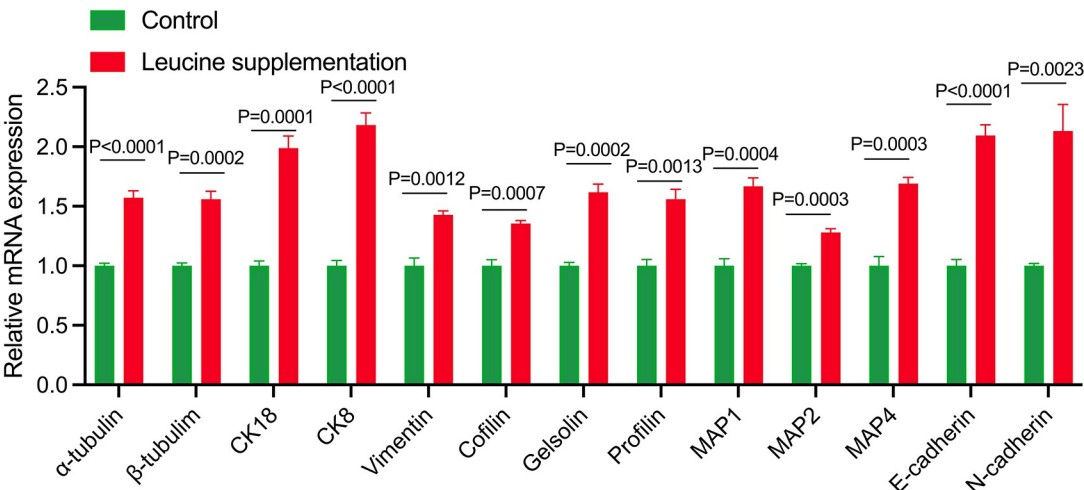

**Fig 6. Verification of cytoskeleton-related genes by QPCR analysis.** To verify the reliability of the results, we selected several key genes of Microtubules (α-tubulin, β-tubulin), Intermediate Filaments (CK18, CK8, Vimentin), Actin-Binding Proteins (Cofilin, Gelsolin, Profilin), Microtubule-Associated Proteins (MAP1, MAP2, MAP4), and Adhesion Proteins (E-cadherin and N-cadherin) for qPCR verification (n = 6).

muscle fibers. Actin filaments are a major component of the thin filaments within myofibrils and are essential for muscle contraction [28]. During muscle contraction, the interaction between actin (thin filaments) and myosin (thick filaments) leads to the sliding of these filaments past each other, resulting in the shortening of the muscle fiber [29]. The cytoskeleton provides the framework that enables this process, known as the sliding filament theory. The cytoskeleton is involved in transmitting the force generated during muscle contraction to the extracellular matrix and ultimately to the tendons and bones [30].

### 3.6. Verification of cytoskeleton-related genes by QPCR analysis

Transcriptome analysis results suggest that the genes mainly regulated by leucine supplementation are enriched in the cytoskeleton pathway. To verify the reliability of the results, we selected several key genes of Microtubules (*α-tubulin*, *β-tubulin*), Intermediate Filaments (*CK18*, *CK8*, *Vimentin)*, Actin-Binding Proteins (*Cofilin*, *Gelsolin*, *Profilin)*, Microtubule-Associated Proteins (*MAP1*, *MAP2*, *MAP4)*, and Adhesion Proteins (E-*cadherin* and *N-cadherin*) for qPCR verification. The qPCR results were consistent with the transcriptome sequencing results, and leucine supplementation could significantly increase the expression of these genes (Fig 6).

## 4. Discussion

### 4.1 The effects of leucine supplementation on the basketball exercise performances

Leucine supplementation resulted in a statistically significant improvement in sprint performance, with times decreasing from 17.4 ± 0.9 to 16.2 ± 0.9 seconds in the leucine group, while the control group exhibited only minimal changes (from 17.3 ± 0.9 to 17.1 ± 0.8 seconds; P = 0.034). This finding suggests that leucine positively influenced speed and agility, likely by promoting muscle recovery, reducing fatigue, and enhancing muscle protein synthesis through the activation of the mTOR signaling pathway. Previous studies have demonstrated that leucine increases mTOR phosphorylation and enhances downstream targets like S6K1

and 4E-BP1, essential for muscle growth and repair, particularly after resistance training [31, 32]. However, those studies primarily focused on strength-based or resistance-trained populations. The current study extends these findings to basketball players, emphasizing leucine's utility in a sport requiring rapid, high-intensity movements. The results align with previous literature but also highlight leucine's unique contribution to sport-specific performance, particularly in activities dependent on fast-twitch muscle fibers and anaerobic energy production.

In terms of three-point shooting accuracy, participants in the leucine group showed a non-significant trend toward improvement, with scores increasing from $10.2 \pm 1.1$ to $11.1 \pm 1.4$, compared to minimal change in the control group (from $10.5 \pm 1.0$ to $10.4 \pm 0.8$; $P > 0.05$). This improvement may reflect subtle enhancements in neuromuscular coordination and muscular endurance, potentially mediated by leucine's role in modulating fatigue and improving recovery. While studies have shown that leucine supports muscle recovery and protein turnover, its effects on technical skills like shooting remain underexplored [33, 34]. The lack of statistical significance in this study may be attributed to the short supplementation duration and limited sample size. Future investigations with extended supplementation periods and larger cohorts could clarify the impact of leucine on precision-based tasks.

Similarly, free-throw performance exhibited a slight, non-significant increase in the leucine group (from $8.0 \pm 0.8$ to $8.4 \pm 0.8$), compared to minimal changes in the control group (from $7.8 \pm 1.2$ to $7.9 \pm 0.7$). This trend may suggest that leucine aids in maintaining consistency and accuracy during repetitive tasks by supporting neuromuscular efficiency and mitigating fatigue. Previous researchers identified leucine as a nutrient signal that activates the mTOR pathway independently of other amino acids, which may contribute to improved muscular endurance [35]. While these findings are promising, the current study does not establish a definitive link between leucine and shooting performance. Additional research incorporating biomechanical analyses and cognitive assessments could further elucidate the mechanisms underlying these trends.

Mental and physical well-being scores improved slightly in the leucine group (from $7.3 \pm 0.9$ to $7.9 \pm 0.7$), with minimal changes in the control group (from $7.1 \pm 0.9$ to $7.3 \pm 0.9$; $P > 0.05$). Although not statistically significant, this finding suggests a potential role for leucine in supporting recovery and reducing exercise-induced fatigue, as previously noted [36, 37]. The subjective nature of well-being assessments limits the conclusiveness of this observation, but it provides valuable insight into the athletes' perceived benefits of leucine supplementation. Previous studies have largely focused on objective measures such as muscle hypertrophy and strength [38]; this study adds a subjective dimension, emphasizing the holistic benefits of leucine in athletic performance.

## 4.2 The effects of leucine supplementation on muscle based on transcriptomic insights

The transcriptomic analysis revealed a substantial number of differentially expressed genes (DEGs) associated with immune response, cytokine production, and neutrophil extracellular trap formation, suggesting that leucine supplementation may also modulate immune and inflammatory responses. This becomes of utmost relevance in athletics, as an appropriate immune response will enable one to have better recovery and functioning muscles [39]. Exercise, especially eccentric exercises, could be injurious to the muscles, known to provoke the inflammatory response [40]. Our findings suggest that leucine may also be useful in modulating this inflammatory response and possibly hastening recovery and improving performance.

Weighted gene co-expression network analysis (WGCNA) identified one gene module related to muscle growth. This has thus underlined further the potential of leucine for the

induction of muscle hypertrophy. The pathway enrichment of DEG in the cytoskeleton underlines the importance of structural proteins to muscle adaptation and growth. QPCR confirmation of several key cytoskeletal genes, including α-tubulin, β-tubulin, further confirmed the transcriptomic findings with CK18, CK8, vimentin, cofilin, gelsolin, profilin, and various microtubule-associated proteins- MAP1, MAP2, and MAP4-and adhesion molecules such as E-cadherin and N-cadherin. These cytoskeletal elements are highly important for muscle cell integrity and function, supporting the notion that leucine supplementation may be acting at a molecular level to enhance muscle strength and resilience.

### 4.3 Limitations and future directions

While the performance improvements in three-point and free-throw accuracy were not statistically significant, they indicated a positive trend that required further investigation. The relatively short duration of supplementation and the limited sample size might have constrained our ability to detect more pronounced effects. Future studies should consider longer durations, larger cohorts, and additional performance measures (e.g., vertical jump, endurance tests) to comprehensively assess leucine's benefits. Furthermore, exploring interactions with other nutrients and training regimens would provide deeper insights into leucine's role in sports nutrition.

### 4.4 Conclusion

These results supported the hypothesis that leucine supplementation could optimize muscle function by enhancing cytoskeletal dynamics, thereby improving athletic performance, particularly in sports that require high-intensity, anaerobic efforts. Given its effects on muscle adaptation and recovery, leucine could also be explored as a supplement to reduce fatigue and accelerate recovery in athletes, promoting sustained performance in both training and competition in the future.

## Supporting information

**S1 File.**
(XLSX)

## Author Contributions

**Conceptualization:** Sinan Wang, Rui Dong.

**Data curation:** Sinan Wang, Weishuai Guo.

**Formal analysis:** Sinan Wang.

**Investigation:** Sinan Wang, Rui Dong.

**Methodology:** Sinan Wang, Weishuai Guo, Rui Dong.

**Project administration:** Rui Dong.

**Resources:** Weishuai Guo.

**Software:** Weishuai Guo, Rui Dong.

**Supervision:** Rui Dong.

**Writing – original draft:** Sinan Wang, Weishuai Guo.

**Writing – review & editing:** Rui Dong.

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
