## [Decision Letter · Decision Letter 0]

20 Sep 2024

PONE-D-24-33132Unraveling the Transcriptomic Effects of Leucine Supplementation on Muscle Growth and Performance in Basketball AthletesPLOS ONE

Dear Dr. Dong,

Thank you for submitting your manuscript to PLOS ONE. After careful consideration, we feel that it has merit but does not fully meet PLOS ONE’s publication criteria as it currently stands. Therefore, we invite you to submit a revised version of the manuscript that addresses the points raised during the review process.

We look forward to receiving your revised manuscript.

Kind regards,

Jinhui Liu

Academic Editor

PLOS ONE

Journal Requirements:

2. We suggest you thoroughly copyedit your manuscript for language usage, spelling, and grammar. If you do not know anyone who can help you do this, you may wish to consider employing a professional scientific editing service. The American Journal Experts (AJE) (https://www.aje.com/) is one such service that has extensive experience helping authors meet PLOS guidelines and can provide language editing, translation, manuscript formatting, and figure formatting to ensure your manuscript meets our submission guidelines. Please note that having the manuscript copyedited by AJE or any other editing services does not guarantee selection for peer review or acceptance for publication. Upon resubmission, please provide the following: ● The name of the colleague or the details of the professional service that edited your manuscript ● A copy of your manuscript showing your changes by either highlighting them or using track changes (uploaded as a *supporting information* file) ● A clean copy of the edited manuscript (uploaded as the new *manuscript* file)

3. In the online submission form, you indicated that “The data underlying the results presented in the study are available from corresponding author upon request.”.

All PLOS journals now require all data underlying the findings described in their manuscript to be freely available to other researchers, either 1. In a public repository, 2. Within the manuscript itself, or 3. Uploaded as supplementary information. This policy applies to all data except where public deposition would breach compliance with the protocol approved by your research ethics board. If your data cannot be made publicly available for ethical or legal reasons (e.g., public availability would compromise patient privacy), please explain your reasons on resubmission and your exemption request will be escalated for approval.

6. Please ensure that you refer to Figure 6 in your text as, if accepted, production will need this reference to link the reader to the figure.

Additional Editor Comments: 

Authors should revise according to the suggestions of reviewers. The modifications should be marked. A point to point response letter is needed.

Reviewers' comments:

Reviewer's Responses to Questions

**Comments to the Author**

1. Is the manuscript technically sound, and do the data support the conclusions?

Reviewer #1: Yes

Reviewer #2: Partly

2. Has the statistical analysis been performed appropriately and rigorously? 

Reviewer #1: Yes

Reviewer #2: I Don't Know

3. Have the authors made all data underlying the findings in their manuscript fully available?

Reviewer #1: Yes

Reviewer #2: No

4. Is the manuscript presented in an intelligible fashion and written in standard English?

Reviewer #1: No

Reviewer #2: No

5. Review Comments to the Author

Reviewer #1: **Linguistically, be sure to have a native English person review the article.

**In any part of the article where we use abbreviations, we must first bring the full vocabulary and then use the abbreviations.

**In the text of the article, you mentioned GAP, but no explanation was given.

**Determine the reruns of the feeding protocol and what the rest of the subjects' diet was like.

**In the discussion, you mentioned that studies, but you mentioned one study.

**Mention the strengths and weaknesses of the work.

Reviewer #2: Thank you for providing me the opportunity of manuscript entitle: “Unraveling the Transcriptomic Effects of Leucine Supplementation on Muscle Growth and Performance in Basketball Athletes”. The title is interesting , however some concerns are raised or following points are suggested:

Major grammatical revisions are required in all parts of the manuscript.

1- Abstract

The following phrase or sentences need rewording:

- ” we measure their exercise performance like 282 feet sprint”

- subject self-rating at practices as baseline

- Subsequently, the same exercise performances were recorded as above

- but there still have promoted tendency

-

doses of leucine need to be included.

Conclusion is not based on the results.

The tense of most sentences should become past

2- introduction

- The physiological rationale selecting variables of sport performance according to physiological base is not clear and must be mentioned.

- The rationale for selecting every gene must be mentioned.

- What is the necessity of this research while the effects of leucine on hypertrophy as well as performance have been investigated in several previous studies.

3- Method

- The method of recruiting participants is not clear.

- The method of sample size calculation is not clear.

- This sentence is not proper for method” The body weight, height and Body mass index (BMI), which accounts for both height and weight, was not significantly different between the 2 groups. “

- Ethics approval is necessary for this manuscript while it was mentioned an N/A.

- The phrase “The measures measured included timed sprints and shooting accuracy,

“ needs rewording

- Tissue sampling method is not available.

- Physical performance tests must be explained in details.

4- Discussion

- All measured variables findings must be disussed individually.

- Limitations must be included.

6. PLOS authors have the option to publish the peer review history of their article (what does this mean?). If published, this will include your full peer review and any attached files.

Reviewer #1: **Yes: **Yaser Alikhajeh

Reviewer #2: No

---

## [Author Response · Author response to Decision Letter 0]

21 Oct 2024

Review Comments to the Author

Reviewer #1: **Linguistically, be sure to have a native English person review the article.

R: Thanks, we have polished the language of this article by a professional English native speaker.

**In any part of the article where we use abbreviations, we must first bring the full vocabulary and then use the abbreviations.

R: Thanks, we have revised these, we have provided the full vocabulary of the abbreviations when they firstly occurred.

**In the text of the article, you mentioned GAP, but no explanation was given.

R: it denotes that while leucine's general benefits for athletic performance are well-documented, there is insufficient or inconclusive data on its specific impact or application to basketball players. Therefore, our research seeks to fill this gap by exploring how leucine supplementation could influence performance metrics, recovery, and muscle function specifically in the context of basketball. We have added the explanation of this in the main text.

**Determine the reruns of the feeding protocol and what the rest of the subjects' diet was like.

R: For the feeding protocol, During the exercise test, all the participants followed a controlled diet consisting of prepackaged meals and snacks that provided 1.5 times their resting energy expenditure (REE). The diet included 1.2 grams of protein per kilogram of body weight per day (g/kg/d), 4 g/kg/d of carbohydrates, and the remainder of calories from fat, ensuring a balanced intake tailored to their energy needs. Participants were instructed to drink only water in addition to the provided meals and snacks, standardizing hydration and nutrient intake throughout the testing period to control for variables that could influence exercise performance.

For the drinking, participants were given a 0.25 g·kg-1 carbohydrate solution mixed with 16 ounces of water (total 0.50 g·kg-1) 30 minutes before and immediately after exercise. The independent variable was the addition of 25 mg·kg-1 leucine powder to the test drinks before and after exercise (total 50 mg·kg-1) for the experimental group.

**In the discussion, you mentioned that studies, but you mentioned one study.

R: Thanks, we have revised this. ‘Studies have highlighted the pivotal role of leucine in stimulating muscle protein synthesis via the mTOR signaling pathway. Crozier et al. (2005) demonstrated that leucine activates mTOR, promoting downstream effectors like S6K1 and 4E-BP1, which enhance translation initiation, crucial for muscle growth and repair. Norton et al. (2009) showed that leucine supplementation after resistance exercise increases phosphorylation of mTOR and its downstream targets, resulting in enhanced protein synthesis, supporting its role in exercise-induced hypertrophy. Additionally, Anthony et al. (2000) found that leucine acts as a nutrient signal to the mTOR pathway, even in the absence of other amino acids, promoting muscle protein synthesis independently of other dietary factors. “

**Mention the strengths and weaknesses of the work.

R: I have added the discussion of the strengths and weaknesses of the work in the discussion part. “However, the study has some limitations, such as the lack of statistically significant improvements in other performance metrics and short duration, which may have restricted its ability to capture the full benefits of leucine supplementation. The focus on short-term effects and the absence of data on leucine’s interactions with other nutrients also limit the broader applicability of the findings.”

Reviewer #2: Thank you for providing me the opportunity of manuscript entitle: “Unraveling the Transcriptomic Effects of Leucine Supplementation on Muscle Growth and Performance in Basketball Athletes”. The title is interesting , however some concerns are raised or following points are suggested:

Major grammatical revisions are required in all parts of the manuscript.

1- Abstract

The following phrase or sentences need rewording:

- ” we measure their exercise performance like 282 feet sprint”

- subject self-rating at practices as baseline

- Subsequently, the same exercise performances were recorded as above

- but there still have promoted tendency

-doses of leucine need to be included.

Conclusion is not based on the results.

The tense of most sentences should become past

R: Thanks for your suggestions, we have revised our abstract according to your advice.

2- introduction

- The physiological rationale selecting variables of sport performance according to physiological base is not clear and must be mentioned.

R: “The selected variables—282 feet sprint, free throws, and three-point field goals—reflect key physiological and skill demands in basketball. The sprint tests anaerobic capacity and fast-twitch muscle performance crucial for quick, high-intensity movements, while free throws and three-point shots measure precision, muscle coordination, and endurance needed for accurate shooting. Together, these variables capture both the physical and technical aspects of basketball performance.” We have included these in the introduction.

- The rationale for selecting every gene must be mentioned.

R: Yes, the rational of selecting genes have been included in the introduction.

- What is the necessity of this research while the effects of leucine on hypertrophy as well as performance have been investigated in several previous studies.

R: Although the general benefits of leucine for athletic performance are well-established, there is a lack of research specifically examining its effects in basketball, particularly in terms of how leucine supplementation may impact performance metrics, recovery, and muscle function within this sport.

3- Method

- The method of recruiting participants is not clear.

R: thanks, we have revised this. "In total, 20 men's basketball players (aged 18–23 years) were recruited from a university basketball team for this study. All participants held a Level 1 basketball athlete qualification. Volunteers were recruited through advertisements. This study received approval from the university ethics committee." 

- The method of sample size calculation is not clear.

R: Thanks, we have revised this, “In this study, data were processed using GraphPad Prism 8.0.1 software. A two-way ANOVA was employed to analyze the data, allowing for the examination of the effects of two independent variables and their interaction on the dependent variable. Following the ANOVA, Tukey’s multiple comparison test was applied to identify specific group differences. A p-value of less than 0.05 was considered statistically significant.”

- This sentence is not proper for method” The body weight, height and Body mass index (BMI), which accounts for both height and weight, was not significantly different between the 2 groups. “

R: We have rewritten this sentence. “The body weight and height of participants were measured to ensure comparability between the two groups. No significant differences were found in these parameters between the groups.”

- Ethics approval is necessary for this manuscript while it was mentioned an N/A.

R: we have received the ethics approval from university ethics committee, the statement has been updated in the main text.

- The phrase “The measures measured included timed sprints and shooting accuracy,

“ needs rewording

R: We have revised this. “The assessments included timed sprints and shooting accuracy.”

- Tissue sampling method is not available.

R: we have revised this. “In this study, muscle samples from the biceps were collected from each group after a 28-day experimental period for RNA sequencing (RNA-seq) analysis. Six replicate muscle samples were obtained from each group. The biceps samples were carefully excised under sterile conditions and immediately flash-frozen in liquid nitrogen to preserve RNA integrity. They were then stored at -80°C until RNA extraction. This process ensured high-quality samples for subsequent RNA-seq analysis.”

- Physical performance tests must be explained in details.

R: Thanks, we have added more details about this in methods section.

4- Discussion

- All measured variables findings must be disussed individually.

R: Thanks, we have revised this, we have discussed all the measure variables findings in the discussion. “ Notably, a significant improvement in the 282-foot sprint performance following leucine supplementation, with sprint times decreasing from 17.4 ± 0.9 to 16.2 ± 0.9 seconds, compared to a minimal change in the control group (P = 0.034). This suggests that leucine may enhance speed and agility. In contrast, while there was a trend towards improvement in three-point shooting accuracy (from 10.2 ± 1.1 to 11.1 ± 1.4) and free throw performance (from 8.0 ± 0.8 to 8.4 ± 0.8) in the leucine group, these changes were not statistically significant compared to the control group. This indicates that while leucine might have a positive effect on shooting performance, the results were not conclusive. Additionally, subjective ratings of mental and physical well-being showed a slight improvement in the leucine group (from 7.3 ± 0.9 to 7.9 ± 0.7), whereas the control group remained relatively stable (from 7.1 ± 0.9 to 7.3 ± 0.9). Although this improvement was not statistically significant (P > 0.05), it also suggests a potential benefit of leucine on overall well-being.”

- Limitations must be included.

R: I have added the limitation of the work in the discussion part. “However, the study has some limitations, such as the lack of statistically significant improvements in other performance metrics and short duration, which may have restricted its ability to capture the full benefits of leucine supplementation. The focus on short-term effects and the absence of data on leucine’s interactions with other nutrients also limit the broader applicability of the findings.”

---

## [Decision Letter · Decision Letter 1]

4 Nov 2024

PONE-D-24-33132R1Unraveling the Transcriptomic Effects of Leucine Supplementation on Muscle Growth and Performance in Basketball AthletesPLOS ONE

Dear Dr. Dong,

Thank you for submitting your manuscript to PLOS ONE. After careful consideration, we feel that it has merit but does not fully meet PLOS ONE’s publication criteria as it currently stands. Therefore, we invite you to submit a revised version of the manuscript that addresses the points raised during the review process.

We look forward to receiving your revised manuscript.

Kind regards,

Jinhui Liu

Academic Editor

PLOS ONE

**Additional Editor Comments:**

Authors should revise according to the suggestions of reviewers. The modifications should be marked. A point to point response letter is needed.

Reviewers' comments:

Reviewer's Responses to Questions

**Comments to the Author**

1. If the authors have adequately addressed your comments raised in a previous round of review and you feel that this manuscript is now acceptable for publication, you may indicate that here to bypass the “Comments to the Author” section, enter your conflict of interest statement in the “Confidential to Editor” section, and submit your "Accept" recommendation.

Reviewer #1: All comments have been addressed

Reviewer #2: (No Response)

2. Is the manuscript technically sound, and do the data support the conclusions?

Reviewer #1: Yes

Reviewer #2: No

3. Has the statistical analysis been performed appropriately and rigorously? 

Reviewer #1: Yes

Reviewer #2: Yes

4. Have the authors made all data underlying the findings in their manuscript fully available?

Reviewer #1: Yes

Reviewer #2: Yes

5. Is the manuscript presented in an intelligible fashion and written in standard English?

Reviewer #1: Yes

Reviewer #2: No

6. Review Comments to the Author

Reviewer #1: (No Response)

Reviewer #2: Thank you for providing me the opportunity of manuscript entitle: “Unraveling the Transcriptomic Effects of Leucine Supplementation on Muscle Growth and Performance in Basketball Athletes”. The title is interesting , however some concerns are raised or following points are suggested:

Still there are Major grammatical problems, and most of required revisions have not been considered.

are required in all parts of the manuscript.

1-Abstract

The following phrase or sentences need rewording:

“ we total recruit 20 basketball player and randomly divided into two groups”

-subject self-rating at practices as baseline

-Subsequently, the same exercise performances were recorded as above

-but there still have promoted tendency

Conclusion is not based on the results.

The tense of most sentences should change past

2-introduction

again following questions have not been answered.

- The physiological rationale selecting variables of sport performance according to physiological base is not clear and must be mentioned. Several sentences have been included without any reference.

-The rationale for selecting every gene must be mentioned.

3-Method

-The method of sample size calculation is not clear.

-Ethic committee number is not available.

-This sentence is not proper properly stated” The body weight, height and Body mass index (BMI), which accounts for both height and weight, was not significantly different between the 2 groups. “

-Ethics approval number is necessary for this manuscript .

-Regarding diet, when did they eat food before testing?

-Did you consider muscle hypertrophy (size as an important mediatory factor for performance)?

4-Discussion

-All measured variables findings must be discussed individually.

-

7. PLOS authors have the option to publish the peer review history of their article (what does this mean?). If published, this will include your full peer review and any attached files.

Reviewer #1: **Yes: **Yaser Alikhajeh

Reviewer #2: No

---

## [Author Response · Author response to Decision Letter 1]

19 Nov 2024

1-Abstract

The following phrase or sentences need rewording:

“ we total recruit 20 basketball player and randomly divided into two groups”

R： we have revised this sentence. “ In this study, a total of 20 basketball players were recruited and randomly divided into two groups.”

-subject self-rating at practices as baseline

-Subsequently, the same exercise performances were recorded as above

R: This sentence has been revised. “Prior to leucine supplementation, baseline exercise performance, including a 282-foot sprint, free throws, three-point field goals, and self-rated practice assessments, was evaluated. Participants were then provided with a functional drink, either with or without leucine supplementation (50 mg/kg body weight), for 28 days. Subsequently, the same exercise performance metrics were reassessed.”

-but there still have promoted tendency

R: we have revised this. “For other exercise performance metrics, no significant differences were observed (P > 0.05); however, a trend toward improvement was noted.”

Conclusion is not based on the results.

R: We have revised the summary part in abstract. “In summary, Leucine supplementation improved exercise performance, significantly reducing sprint times and showing trends of improvement in three-point field goals, free throws, and self-rated well-being. Transcriptome analysis identified 3,658 differentially expressed genes enriched in pathways related to immune response, cytokine production, and cell adhesion. Weighted Gene Co-expression Network Analysis (WGCNA) highlighted a key module strongly associated with muscle growth traits, with functional enrichment in cytoskeletal pathways. qPCR validation confirmed upregulation of cytoskeleton-related genes, supporting transcriptomic findings. These results suggest that leucine enhances muscle adaptation by regulating cytoskeletal dynamics, offering molecular insights into its role in improving athletic performance.”

The tense of most sentences should change past

R: the tense of the sentences have been checked.

2-introduction

again following questions have not been answered.

- The physiological rationale selecting variables of sport performance according to physiological base is not clear and must be mentioned. Several sentences have been included without any reference.

R: Thanks, The following sentences have declared the physiological rational of selecting sport performance. “ The selected performance variables—282-foot sprint, free throws, and three-point field goals—were chosen based on their relevance to the physiological and skill demands of basketball. The sprint test was designed to assess anaerobic capacity, which is critical for high-intensity, short-duration efforts, as well as the activation and performance of fast-twitch muscle fibers required for rapid and explosive movements during gameplay. Free throws and three-point shots were included to evaluate precision, muscle coordination, and endurance, all of which are necessary for maintaining shooting accuracy under physical and mental fatigue.”

We have included reference in the miantext as well, thanks for suggestions.

-The rationale for selecting every gene must be mentioned.

R: The following sentences have declare the rationale of selected genes. And we have added the references. “The genes of interest were selected based on their critical roles in maintaining cytoskeletal integrity, regulating muscle contraction, and promoting tissue repair, all of which are essential for muscle adaptation to physical stress and high-intensity activity. Specifically, α-tubulin and β-tubulin are key components of microtubules, which provide structural support and facilitate intracellular transport. CK18 and CK8, as intermediate filaments, contribute to cellular integrity and resilience under mechanical stress. Vimentin is involved in maintaining cell shape and stability during dynamic cellular processes. Actin-binding proteins, such as cofilin, gelsolin, and profilin, regulate actin filament assembly and disassembly, crucial for muscle contraction and cytoskeletal remodeling. MAP1, MAP2, and MAP4 are microtubule-associated proteins that modulate microtubule stability and dynamics, supporting cellular structure during muscle activity. E-cadherin and N-cadherin are adhesion proteins that play pivotal roles in cell-cell adhesion, necessary for tissue integrity and coordinated muscle function. These genes collectively represent fundamental processes such as actin and microtubule dynamics, cell adhesion, and cytoskeletal stability, which leucine supplementation may influence to enhance muscle strength, function, and recovery.”

3-Method

-The method of sample size calculation is not clear.

R: Thanks, we have declared this. “ The sample size for this study was determined using a power analysis to ensure sufficient statistical power for detecting changes in performance metrics (e.g., sprint times, free-throw accuracy) between the experimental and control groups. A two-tailed analysis with a power of 80% (β = 0.2) and a significance level of 0.05 (α = 0.05) was conducted.”

-Ethic committee number is not available.

R: This experiment has been approved by the Ethics Committee of South China Normal University (SCNU-SPT-2023-031).

-This sentence is not proper properly stated” The body weight, height and Body mass index (BMI), which accounts for both height and weight, was not significantly different between the 2 groups. “

R: We have revised this. “ Participants' body weight and height were measured to ensure comparability between the two groups, and no significant differences were detected.”

-Ethics approval number is necessary for this manuscript .

R: This experiment has been approved by the Ethics Committee of South China Normal University (SCNU-SPT-2023-031).

-Regarding diet, when did they eat food before testing?

R: “Participants consumed their last prepackaged meal or snack approximately 2-3 hours before each exercise test to ensure adequate digestion and energy availability while minimizing the risk of gastrointestinal discomfort during performance assessments. This timing was standardized across all participants to maintain consistency.” We have included this in the Methods.

-Did you consider muscle hypertrophy (size as an important mediatory factor for performance)?

R: Muscle hypertrophy was not directly measured in this study; however, its potential role as a mediatory factor for basketball performance was considered. Instead of focusing on muscle size, we emphasized functional outcomes, such as sprint performance, shooting accuracy, and subjective well-being, as these are more directly linked to basketball performance. Additionally, the transcriptomic analysis highlighted cytoskeletal and contractile pathways, which are integral to muscle adaptation and function, suggesting an indirect impact on hypertrophy. Future studies could incorporate direct measures of muscle size, such as imaging or biopsy, to better understand its mediatory role in basketball performance.

4-Discussion

-All measured variables findings must be discussed individually.

We appreciate the reviewer’s valuable feedback and agree that discussing each measured variable individually will enhance the clarity and depth of our findings. Below, we provide a detailed discussion of all measured variables, incorporating both performance and molecular insights:

---

## [Decision Letter · Decision Letter 2]

1 Dec 2024

PONE-D-24-33132R2Unraveling the Transcriptomic Effects of Leucine Supplementation on Muscle Growth and Performance in Basketball AthletesPLOS ONE

Dear Dr. Dong,

Thank you for submitting your manuscript to PLOS ONE. After careful consideration, we feel that it has merit but does not fully meet PLOS ONE’s publication criteria as it currently stands. Therefore, we invite you to submit a revised version of the manuscript that addresses the points raised during the review process.

We look forward to receiving your revised manuscript.

Kind regards,

Jinhui Liu

Academic Editor

PLOS ONE

Journal Requirements:

Reviewers' comments:

Reviewer's Responses to Questions

**Comments to the Author**

1. If the authors have adequately addressed your comments raised in a previous round of review and you feel that this manuscript is now acceptable for publication, you may indicate that here to bypass the “Comments to the Author” section, enter your conflict of interest statement in the “Confidential to Editor” section, and submit your "Accept" recommendation.

Reviewer #2: All comments have been addressed

2. Is the manuscript technically sound, and do the data support the conclusions?

Reviewer #2: Yes

3. Has the statistical analysis been performed appropriately and rigorously? 

Reviewer #2: Yes

4. Have the authors made all data underlying the findings in their manuscript fully available?

Reviewer #2: Yes

5. Is the manuscript presented in an intelligible fashion and written in standard English?

Reviewer #2: Yes

6. Review Comments to the Author

Reviewer #2: please consider minor grammatical revisions.

Note: please look at the conclusion, is it a conclusion of your manuscript?

4.4 Conclusion

We have now thoroughly discussed each measured variable and its

implications for athletic performance and molecular adaptations. By

addressing these findings individually, we aim to provide a clearer

understanding of leucine’s potential benefits and limitations within the context

of basketball performance. We hope this revision meets the reviewer’s

expectations and clarifies our study’s contributions.

7. PLOS authors have the option to publish the peer review history of their article (what does this mean?). If published, this will include your full peer review and any attached files.

Reviewer #2: No

---

## [Author Response · Author response to Decision Letter 2]

4 Dec 2024

Journal Requirements:

R:Dear editor, we have double-checked the references and make a revision.

Reviewer #2: please consider minor grammatical revisions.

Note: please look at the conclusion, is it a conclusion of your manuscript?

4.4 Conclusion

We have now thoroughly discussed each measured variable and its

implications for athletic performance and molecular adaptations. By

addressing these findings individually, we aim to provide a clearer

understanding of leucine’s potential benefits and limitations within the context

of basketball performance. We hope this revision meets the reviewer’s

expectations and clarifies our study’s contributions.

R: Sorry, its a mistake. We have revised the conclusion as below. “4.4 Conclusion

These results supported the hypothesis that leucine supplementation could optimize muscle function by enhancing cytoskeletal dynamics, thereby improving athletic performance, particularly in sports that require high-intensity, anaerobic efforts. Given its effects on muscle adaptation and recovery, leucine could also be explored as a supplement to reduce fatigue and accelerate recovery in athletes, promoting sustained performance in both training and competition in the future.

---

## [Decision Letter · Decision Letter 3]

15 Dec 2024

Unraveling the Transcriptomic Effects of Leucine Supplementation on Muscle Growth and Performance in Basketball Athletes

PONE-D-24-33132R3

Dear Dr. Dong,

We’re pleased to inform you that your manuscript has been judged scientifically suitable for publication and will be formally accepted for publication once it meets all outstanding technical requirements.

Kind regards,

Jinhui Liu

Academic Editor

PLOS ONE

Additional Editor Comments (optional):

The authors have addressed the reviewers' concerns properly and revised the manuscript accordingly. The manuscript can be accepted for publication in its current form

Reviewers' comments:

Reviewer's Responses to Questions

**Comments to the Author**

1. If the authors have adequately addressed your comments raised in a previous round of review and you feel that this manuscript is now acceptable for publication, you may indicate that here to bypass the “Comments to the Author” section, enter your conflict of interest statement in the “Confidential to Editor” section, and submit your "Accept" recommendation.

Reviewer #2: All comments have been addressed

2. Is the manuscript technically sound, and do the data support the conclusions?

Reviewer #2: Yes

3. Has the statistical analysis been performed appropriately and rigorously? 

Reviewer #2: Yes

4. Have the authors made all data underlying the findings in their manuscript fully available?

Reviewer #2: Yes

5. Is the manuscript presented in an intelligible fashion and written in standard English?

Reviewer #2: Yes

6. Review Comments to the Author

Reviewer #2: Thank you. recommendations have been considered. Best wishes for your success in future research and academic study .

7. PLOS authors have the option to publish the peer review history of their article (what does this mean?). If published, this will include your full peer review and any attached files.

Reviewer #2: No

---

## [Editor Report · Acceptance letter]

19 Dec 2024

PONE-D-24-33132R3 

PLOS ONE

Dear Dr. Dong, 

I'm pleased to inform you that your manuscript has been deemed suitable for publication in PLOS ONE. Congratulations! Your manuscript is now being handed over to our production team.

Kind regards, 

on behalf of

Dr. Jinhui Liu 

Academic Editor

PLOS ONE